# MITF activity is regulated by a direct interaction with RAF proteins in melanoma cells

Charlène Estrada[1,2,3,4,5], Liliana Mirabal-Ortega [1,2,3,4,5,6], Laurence Méry[1,2,3,4,5,6], Florent Dingli [7], Laetitia Besse[8,9], Cedric Messaoudi [8,9], Damarys Loew [7], Celio Pouponnot [1,2,3,4,5,6], Corine Bertolotto[10], Alain Eychène [1,2,3,4,5,6] & Sabine Druillennec [1,2,3,4,5,6✉]

The MITF transcription factor and the RAS/RAF/MEK/ERK pathway are two interconnected main players in melanoma. Understanding how MITF activity is regulated represents a key question since its dynamic modulation is involved in the phenotypic plasticity of melanoma cells and their resistance to therapy. By investigating the role of ARAF in NRAS-driven mouse melanoma through mass spectrometry experiments followed by a functional siRNA-based screen, we unexpectedly identified MITF as a direct ARAF partner. Interestingly, this interaction is conserved among the RAF protein kinase family since BRAF/MITF and CRAF/MITF complexes were also observed in the cytosol of NRAS-mutated mouse melanoma cells. The interaction occurs through the kinase domain of RAF proteins. Importantly, endogenous BRAF/MITF complexes were also detected in BRAF-mutated human melanoma cells. RAF/MITF complexes modulate MITF nuclear localization by inducing an accumulation of MITF in the cytoplasm, thus negatively controlling its transcriptional activity. Taken together, our study highlights a new level of regulation between two major mediators of melanoma progression, MITF and the MAPK/ERK pathway, which appears more complex than previously anticipated.

[1] Institut Curie, Centre de Recherche, F-91405 Orsay, France. [2] INSERM U1021, Centre Universitaire, F-91405 Orsay, France. [3] CNRS UMR 3347, Centre Universitaire, F-91405 Orsay, France. [4] Université Paris-Saclay, F-91405 Orsay, France. [5] PSL Research University, F-75006 Paris, France. [6] Equipe Labellisée Ligue Nationale Contre le Cancer, F-91405 Orsay, France. [7] Institut Curie, PSL Research University, Centre de Recherche, Laboratoire de Spectrométrie de Masse Protéomique, 26 rue d'Ulm, 75248 Paris, Cedex05France. [8] Institut Curie, PSL Research University, CNRS UMS 2016, F-91401 Orsay, France. [9] Université Paris-Saclay, INSERM US43, F-91401 Orsay, France. [10] Université Côte d'Azur, INSERM U1065, Centre Méditerranéen de Médecine Moléculaire (C3M), Nice, France. ✉email: sabine.druillennec-rodiere@curie.fr

Cutaneous melanoma is an aggressive tumor arising from malignant transformation of melanocytes[1]. The RAS/RAF/ MEK/ERK is a key signaling pathway frequently mutated in cutaneous melanoma since activating mutations in either *NRAS* or *BRAF* genes occur in 15–20% and 40–50% of cases, respectively, the two main mutations being NRAS$^{Q61K}$ and BRAF$^{V600E}$[2,3]. RAS is a GTPase activated via membrane-bound receptors upon stimulation by growth factors. In its GTP-bound form, RAS recruits effectors at the membrane and stimulates a number of downstream intracellular signaling pathways, including the MAPK/ERK pathway[4]. The three RAF serine-threonine kinases, ARAF, BRAF and CRAF, conserved in vertebrates are among the main RAS effectors. RAF activation enables subsequent activation by phosphorylation of MEK1 and MEK2, which in turn activate ERK1 and ERK2[5]. Once activated, ERK phosphorylates cytoplasmic substrates and regulates a wide variety of transcription programs when translocated into the nucleus, thus leading to modulation of key biological processes such as cell proliferation, survival, migration or differentiation[6].

Using conditional knockout of BRAF and/or CRAF in a mouse melanoma model induced by NRAS$^{Q61K}$, we showed that while BRAF is required downstream of activated NRAS for tumor initiation, both BRAF and CRAF play compensatory functions during late phases of melanomagenesis, thus highlighting the addiction of melanoma to the RAF/ERK pathway[7]. Interestingly, we demonstrated that in the absence of BRAF and CRAF, ARAF alone can sustain both ERK activation and proliferation in NRAS-mutated melanoma cells. In this context, we also observed that ARAF homodimers are sufficient to induce ERK paradoxical activation by Vemurafenib, an inhibitor of BRAF$^{V600E}$ kinase activity widely used in clinics. Our results suggested a dependency toward ARAF kinase, as well as a possible role of ARAF in resistance mechanisms in cutaneous melanoma. The potential role of ARAF in NRAS-induced melanoma was further strengthened by an *in silico* search in public databases that allowed to identify patients with metastatic melanomas harboring an ARAF mutation associated with activating NRAS mutations[7]. Moreover, these observations have recently gained credit with the identification of recurrent activating ARAF mutations in melanoma patients resistant to Belvarafenib, a RAF dimer inhibitor[8]. Nevertheless, ARAF remains the least studied member of the RAF family because: (i) ARAF displays the lowest kinase activity towards MEK compared to other RAF proteins[9], (ii) in most cellular models, the role of ARAF is hidden by the predominant roles of BRAF and CRAF.

Microphtalmia-associated transcription factor (MITF) is a master regulator of the melanocytic lineage since it is essential for the differentiation, survival and proliferation of melanocytes[10,11]. MITF belongs to the MiT family, gathering bHLH-LZ domain transcription factors (TFEB, TFEC and TFE3), that can homo- or hetero-dimerize to regulate gene expression[12]. Expressed in about 80% of human melanoma[13,14], MITF displays a central regulatory role in melanoma cell phenotypic plasticity. A proposed rheostat model suggests that the global level of MITF activity correlates with the phenotype of melanoma cells: at high levels of activity, MITF sustains the proliferative state of melanoma cells while at lower levels, MITF is associated with an invasive and stem-like phenotype[15–18]. In line with its central role, MITF is finely regulated to ensure the homeostasis of melanocytes or melanoma cells[11]. Among its numerous post-translational regulators, MITF is regulated by ERK2, that phosphorylates the S73 residue inducing both proteasome-mediated degradation and increased activity *via* the recruitment of p300/CBP transcription cofactor[19–23]. Altogether, the post-translational regulation of MITF by ERK pathway has complex consequences regarding MITF activity, depending on cellular context[14].

To better characterize the role of ARAF in NRAS-driven melanoma, we searched for new ARAF interactors by mass spectrometry. Our results showed that ARAF directly binds to the transcription factor MITF. ARAF/MITF complexes were found in the cytosol of NRAS-mutated mouse melanoma cells. Not only ARAF, but also BRAF and CRAF interacted with MITF. Importantly, endogenous BRAF/MITF complexes were also evidenced in BRAF-mutated human melanoma cells, thus emphasizing the conservation of RAF/MITF interaction in human. At the functional level, RAF/MITF interaction modulates MITF nuclear localization, thus regulating its transcriptional activity. Taken together, these results highlight a new level of regulation of MITF by RAF, two key players of melanoma biology.

## Results and discussion

**Identification of new ARAF partners by large-scale analysis.** Although our knowledge of ARAF kinase has enlarged over the last decade[24], ARAF remains understudied compared to the other members of RAF family. Owing to the redundant roles and the high homology of RAF kinases as well as the weak kinase activity of ARAF compared to BRAF and CRAF, it is challenging to study ARAF function in most cellular models where BRAF and CRAF are also expressed. In addition, attribution of a specific function to each RAF kinase is further hampered by their propensity to heterodimerize, especially when looking for binding partners. In the present study, we took advantage of a genetically engineered NRAS-driven melanoma mouse model allowing concomitant ablation of BRAF and CRAF to investigate the role of ARAF. Tumour cells derived from these mice constitute a well-adapted model to study ARAF function in the melanoma context in absence of BRAF and CRAF expression (ARAF-only cells)[7]. The ARAF interactome was established by immunoprecipitation of the endogenous ARAF protein from ARAF-only compared to control cells followed by analysis of the immune complexes by mass spectrometry in label-free conditions (Fig. 1). ARAF-only cells, which emerged after *braf* and *craf* genes ablation in melanoma cultures established from primary NRAS-induced tumors, are highly dependent on ARAF expression for their growth and survival[7]. Control cells display normal levels of BRAF and CRAF, but express low level of ARAF due to shRNA-mediated knockdown, thus allowing relative quantification of the data. The distribution of the 2700 ARAF-interacting proteins in ARAF-only or control cells (Supplementary Data 1) is illustrated by the volcano plot (Fig. 1a). We performed bioinformatics analysis using a subset of 431 interactors enriched in ARAF only cells and selected as follows: proteins with number of peptides≥9, ratio>2 and adjusted *p* value < 0.001 and proteins exclusively identified in ARAF-only cells (359 and 72 proteins, respectively) (Supplementary Data 2). KEGG pathway visualization revealed an enrichment in the MAPK signaling pathway, in particular MEK1, the ARAF direct downstream substrate, thus strenghtening the reliability of the experimental approach (Supplementary Fig. 1). Of note, NRAS, the direct upstream interactor of ARAF, is also found in the interactome but did not reach all the cut-offs. While the number of peptides and ratio were correct (peptides = 13, ratio = 12), the adjusted *p* value = 0.009 was above the selected cut-off. In addition, several 14-3-3 proteins (*Ywhab* and *Ywhaz* coding genes) are present in the 431 ARAF interactors subset. Although not included in the KEGG maps (Supplementary Fig. 1), 14-3-3 proteins are also involved in MAPK signaling by directly binding and regulating RAF kinases. We also performed process and pathway enrichment analysis as well as Protein-Protein Interaction (PPI) enrichment analysis with Metascape[25]. The most enriched process was related to Rho GTPases signaling (Supplementary Data 2). PPI enrichment analysis confirmed the

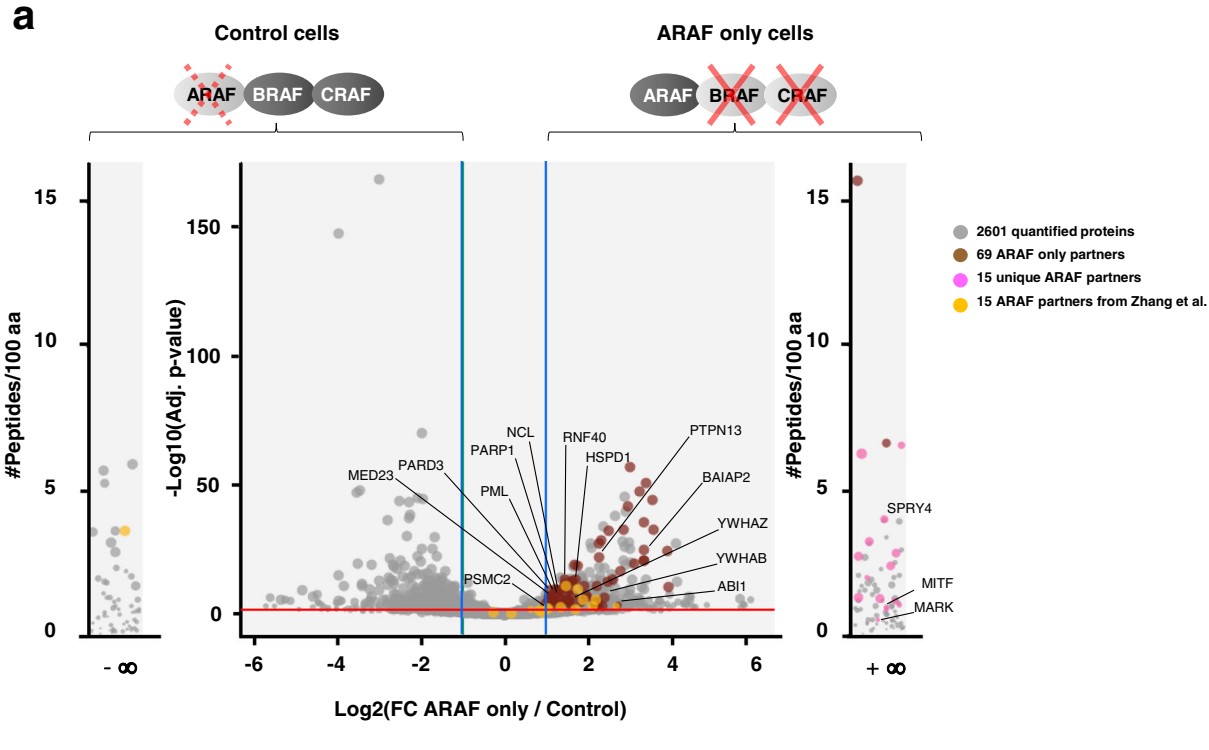

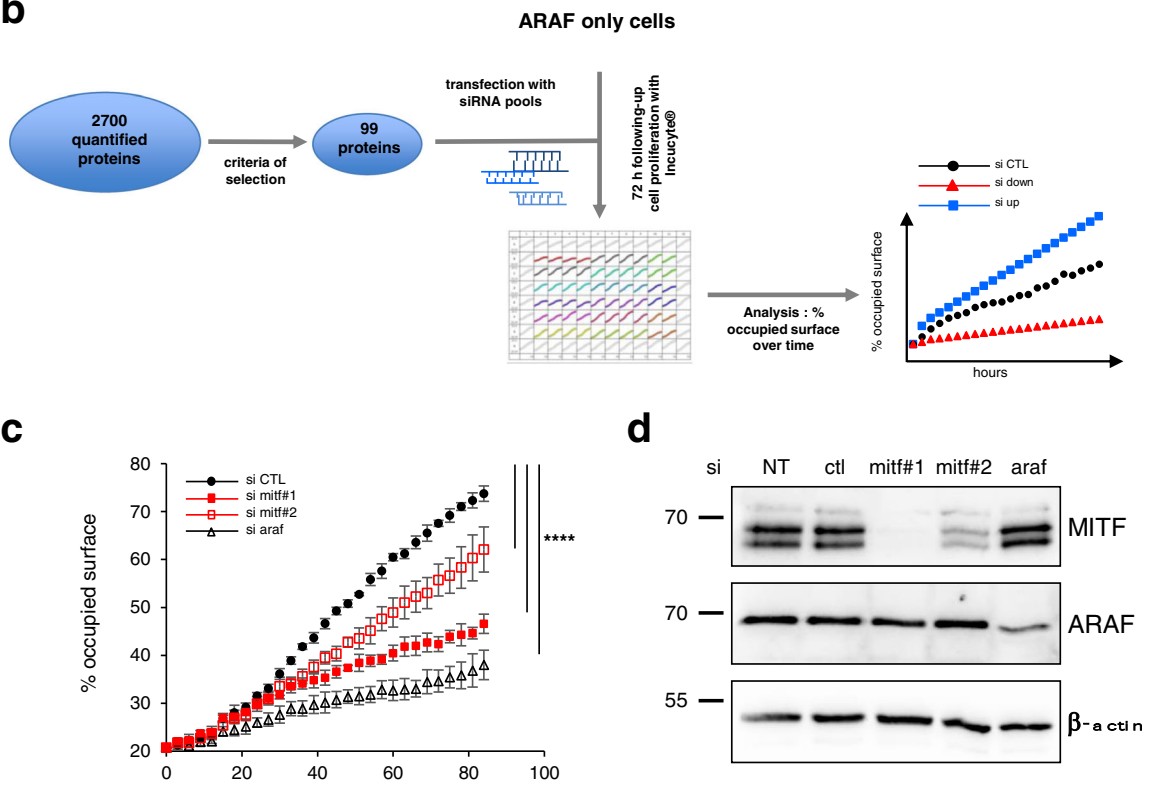

enrichment in Rho GTPases signaling network, but also highlighted mitochondrial processes, such as TCA cycle and respiratory electron transport, mitochondrial translation or fatty acid beta-oxidation (Supplementary Fig. 2). These observations are in agreement with previous published data. Indeed, among RAF family members, it has been described that CRAF regulates Rho signaling independently of its kinase activity by interacting and controlling the subcellular localization of Rok-α[26,27]. Moreover, RAF proteins can be found localized at the mitochondria where they play a role in apoptotic processes or modulate metabolic cell activity[28]. ARAF and CRAF regulate apoptosis by interacting with apoptotic factors[29–31] and the activated form of BRAF has been found localized at the mitochondria where it regulates oxidative metabolism[32].

**Fig. 1 Identification of MITF as an ARAF partner. a** Volcano plot representation of ARAF-binding proteins identified by proteomic analysis. Endogenous ARAF was immunoprecipitated from ARAF-only or control cells lysates. As indicated, ARAF-only cells are double knockout for BRAF and CRAF. Control cells display normal levels of BRAF and CRAF and low levels of ARAF. Binding partners were obtained by using quantitative label-free mass spectrometry analysis performed from five ARAF-only and four control cells replicates. The volcano plot represents the 2700 quantified proteins in control and ARAF-only cells with X axis indicating the log2 fold change (FC) (ARAF-only *versus* control cells) and Y axis the -log10 of adjusted *p* value. The non-axial vertical lines (in blue) denote absolute fold change of 2 while the non-axial horizontal line (in red) denotes the adjusted *p* value of ratio significance of 0.001. External plots show unique proteins with peptides identified only in one sample type (left in control and right in ARAF-only cells). **b** Schematic representation of the workflow to identify new relevant ARAF partners. The 99 proteins selected for further analysis through a siRNA based screen (Supplementary Data 3) are indicated in panel **a**: 69 proteins enriched in ARAF-only cells labeled in brown, 15 unique partners in ARAF-only cells in pink and 15 ARAF interactors published by Zhang et al.[33] in orange. ARAF-only cells proliferation was measured during 72 h by using IncuCyte® technology after transfection with siRNA pools targeting each of the 99 putative partners (Supplementary Fig. 3). The theorical curve shows the percentage of occupied surface over time for a given knockdown target. SiRNA having a pro-proliferative (si up) or anti-proliferative (si down) effect compared to a negative control (si CTL, black circles) are highlighted with blue squares and red triangles, respectively. **c** Proliferation of ARAF-only cells after transfection with a control siRNA (siCTL, black circles), individual siRNA against MITF (siMITF #1 or siMITF #2, in red squares) or siRNA pool against ARAF (siARAF, open black triangles). Data are the mean ± SD of four replicates (*n* = 4). **** *p* value < 0.0001 compared by a two-way ANOVA with Dunnett's multiple comparisons test. **d** Western blot analysis of MITF and ARAF protein levels in ARAF-only cells non-transfected (NT) or transfected with either siCTL, siMITF, or siARAF. β-actin is used as a loading control.

In order to identify ARAF relevant partners, which functionally impact melanoma cell proliferation, we developed a siRNA-based functional screen on 99 selected targets (Fig. 1b). These 99 interactors were selected as follows: 69 were chosen among the previously described 359 proteins enriched in ARAF-only cells and 15 were from the 72 proteins exclusively identified in ARAF-only cells (Supplementary Data 3). We also included 15 proteins that were found both in our current dataset and in the ARAF interactome published by Zhang et al.[33]. ARAF-only cells growth was followed upon knockdown of the selected partners by siRNA pools transfection (Supplementary Fig. 3). Among the 99 partners tested, 16 impacted the growth of ARAF only melanoma cells. It appeared that 11 ARAF partners had an anti-proliferative effect while 5 proteins were pro-proliferative (labeled in blue and red, respectively in Supplementary Data 3). Among the 132 interactors identified in the ARAF proteome by Zhang et al.[33,34], 107 were commonly found in our dataset, showing the robustness of the approach. Twenty-four of the common identified partners were included in our screen: 9 were selected by the previously described parameters and 15 additional were chosen after bibliographic analysis. It should be noticed that Zhang *et al.* have validated their interactome by coimmunoprecipitation experiments on 12 out of 13 randomly selected ARAF interacting proteins with different functions[33,34]. Eight of these confirmed ARAF partners were tested in the siRNA-based screen and three appear to play a functional role in ARAF only cells: NCL, PARP and PSMC2.

Among the 16 partners that impact melanoma proliferation, we decided to focus on MITF since it represents a key transcription factor for melanoma progression that can be involved in therapy-resistance mechanism. It is also well known to be regulated by the MAPK/ERK pathway[11,14]. Of note, the ARAF interactome by Zhang et al. could not identify MITF as an ARAF partner since it was performed on a heterologous model overexpressing tagged ARAF in HEK293T cells that do not express MITF[33,34]. We next confirmed the pro-proliferative effect of MITF in ARAF-only cells by using two distinct siRNA against MITF in comparison to control siRNA. Since we previously demonstrated that ARAF-only cells rely on ARAF for their proliferation, we included a siRNA targeting ARAF as a positive control (Fig. 1c). Both siRNA against MITF decreased the growth of ARAF-only melanoma cells. Moreover, we observed a good correlation between the effect on cell proliferation and the level of extinction of MITF expression induced by the different siRNA (Fig. 1d), demonstrating that MITF is required for ARAF-only cells growth. We also showed that MITF is required in NRAS-

mutated mouse melanoma cells expressing normal levels of all RAF kinases (Supplementary Fig. 4a, b).

**ARAF directly interacts with MITF.** The interaction between ARAF and MITF was confirmed by coimmunoprecipitation experiments of endogenous proteins in ARAF-only cells (Fig. 2a). As shown in Fig. 2b, endogenous ARAF/MITF complexes were also detected by Proximity Ligation Assay (PLA) in ARAF-only cells, further revealing that this interaction occurred in the cytoplasm of melanoma cells. Importantly, this interaction appeared to be direct since complex formation was observed between ARAF and MITF human purified recombinant proteins, in an in vitro coimmunoprecipitation assay (Fig. 2c).

**Characterization of the RAF/MITF interaction.** While the connection between the ERK/MAPK pathway and MITF is well established in melanoma[14], a direct interaction between RAF kinases and MITF has never been previously demonstrated. We tested whether this interaction was specific of ARAF or shared by all the RAF kinases. HEK293T cells were cotransfected with MITF and each of the three different HA-tagged RAF proteins. Anti-HA immune complexes were then probed with an anti-MITF antibody. Interestingly, we observed that MITF could interact not only with ARAF but also with BRAF and CRAF, the two other members of the RAF family (Fig. 3a–c). PLA experiments in NRAS-mutated murine melanoma cells confirmed the existence of endogenous BRAF/MITF and CRAF/MITF complexes located in the cytoplasm (Supplementary Fig. 4c and d, respectively). This is the first evidence of a direct interaction between RAF kinases family and MITF, two key players in melanoma cell biology. Although an MITF interactome has been previously reported, RAF kinases were not identified in this study since the authors focused specifically on nuclear interactors by performing nuclear purification[35]. The MAPK/ERK pathway being dysregulated by NRAS, but also BRAF mutations in melanoma, we investigated the ability of MITF to interact with the constitutively active BRAF$^{V600E}$ mutant (Fig. 3d). This is the most frequent BRAF mutation in human cancers, which is highly prevalent in melanoma and which markedly increases BRAF kinase activity[36]. We observed that MITF strongly interacts with BRAF$^{V600E}$ with an increased affinity compared to wild type BRAF. To evaluate the requirement of the RAF kinase activity, we also tested the interaction with the BRAF$^{K483M}$ kinase-dead mutant (BRAF$^{KD}$) containing a Lys-to-Met substitution in its kinase domain (Fig. 3d). In contrast to BRAF$^{V600E}$, the capacity

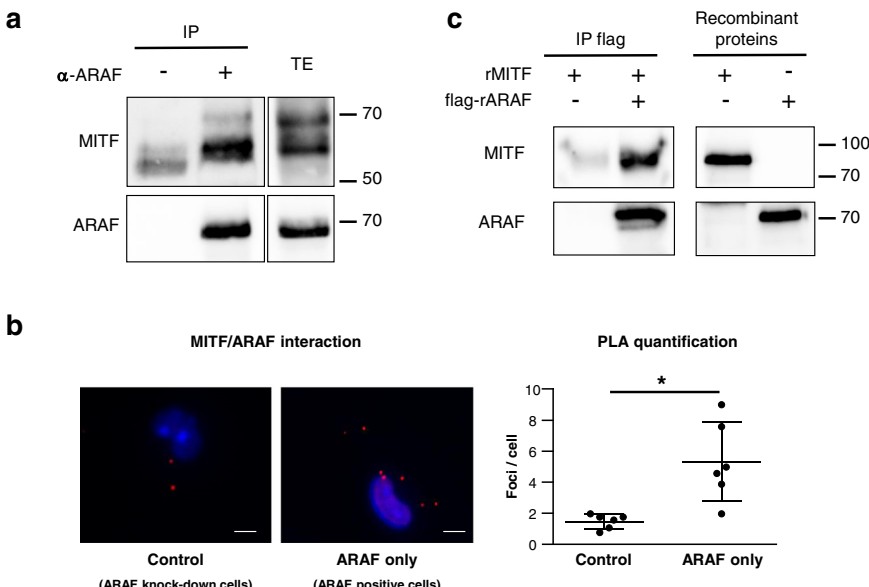

**Fig. 2 Validation of the ARAF/MITF interaction. a** Identification of endogenous ARAF/MITF complexes by co-immunoprecipitation. ARAF-only cells extracts were immunoprecipitated with an anti-ARAF antibody. Immune complexes (IP) and total extracts (TE) were immunoblotted with anti-MITF and anti-ARAF antibodies. **b** Identification of endogenous ARAF/MITF complexes by Proximity Ligation Assay. ARAF/MITF complexes were visualized as red dots in ARAF-only cells compared to control cells expressing an shRNA against ARAF by using a fluorescence microscope. Cell nuclei were stained with DAPI. Scatter plots show the average number of dots per cell (at least 148 nuclei were observed) from six microscopic fields and are representative of three different experiments. Means with standard deviations are shown. *$p$ value = 0.0019 compared by unpaired $t$ test with Welch's correction. Representative pictures from three independent experiments are shown. Scale bar: 100 µm. **c** Identification of a direct ARAF/MITF interaction by in vitro coimmunoprecipitation. Recombinant MITF and flag-tagged ARAF proteins were incubated in NP40 Buffer. ARAF was immunoprecipitated with an anti-flag antibody. Immune complexes and recombinant proteins were blotted with anti-MITF and anti-ARAF antibodies.

of BRAF$^{KD}$ to bind to MITF was decreased as compared to wild type BRAF. Therefore, the strength of the binding directly correlates with the activation state of the RAF proteins since MITF strongly interacts with the activated mutant of BRAF, and much less with the BRAF kinase-dead mutant. These results suggest that not only an active form of the RAF kinase could be required to allow the interaction with MITF, but also that the MITF/BRAF complex formation can occur in a BRAF-mutated context.

We next investigated the role of the different domains of BRAF in the interaction with MITF, by using truncated forms of the protein (Fig. 3e). HEK293T cells were cotransfected with plasmids encoding MITF and either the C-terminus or N-terminus part of BRAF, or both. Of note, it was previously demonstrated that the N-terminus regulatory domain of RAF proteins binds to their C-terminus kinase domain in order to regulate their activity[37]. Accordingly, N- and C-terminus parts co-precipitated when co-expressed (Fig. 3e). Following C-terminus immunoprecipitation in the absence of the N-terminus, a strong interaction with MITF was observed indicating that the N-terminal part of BRAF is not required for MITF binding. Moreover, in these conditions, the presence of the N-terminus did not modify the interaction between MITF and the C-terminus (Fig. 3e, left panel). On the opposite, a weak interaction with MITF was seen when the N-terminal domain was immunoprecipitated in the absence of the C-terminus (Fig. 3e, right panel). However, complex formation between the N-terminus and MITF was strongly increased in the presence of the C-terminal part suggesting that, in this condition, the N-terminus does not interact directly with MITF but through the C-terminal domain. Altogether, the results indicate that complex formation with MITF involves the C-terminus region of RAF proteins that contains the kinase domain. These observations also suggest the requirement of a functional kinase domain to stabilize the interaction between RAF and MITF.

**Detection of endogenous RAF/MITF complexes in BRAF-mutated human melanoma cells.** To further substantiate our observations made in mouse melanoma, we evaluated the formation of RAF/MITF complex in three BRAF-mutated human melanoma cells. As shown in Fig. 4a–c, after immunoprecipitation of endogenous BRAF, we were able to detect MITF in all three different cell lines tested, thus confirming that the BRAF/MITF interaction is conserved in human. Since we observed a correlation between the kinase activity of RAF kinases and their binding to MITF (Fig. 3), BRAF-mutated human melanoma cells were treated with Vemurafenib, an ATP-competitive kinase inhibitor of BRAF. The inhibition of MAPK pathway following Vemurafenib treatment was confirmed by the decrease of ERK phosphorylation (Fig. 4d–f). In these conditions, we observed a slight decrease in the ability of MITF to interact with mutated BRAF compared to untreated cells. This demonstrates that the active site of RAF kinases is not the MITF binding domain and that the interaction does not require kinase activity. The lack of kinase activity requirement is also supported by results in Fig. 3e (left panel) showing that the N-terminus, known to decrease C-terminus kinase activity[37–39], did not modify complex formation between the C-terminus and MITF. Accordingly, no phosphorylation sites or consensus for phosphorylation by RAF kinases, indicating that MITF could be a direct RAF substrate, have been reported. This also suggests that the ability of RAF kinases to bind MITF is not linked to a fully functional kinase active site, but rather due to conformational aspects.

**Functional role of RAF/MITF interaction.** We next investigated how the MITF/RAF complex formation could affect the respective subcellular localization of each partner, knowing that RAF kinases are cytosolic, whereas MITF can shuttle between the cytosol and the nucleus[23]. HEK293T cells were transfected with epitope-tagged MITF and ARAF, BRAF or CRAF and subcellular

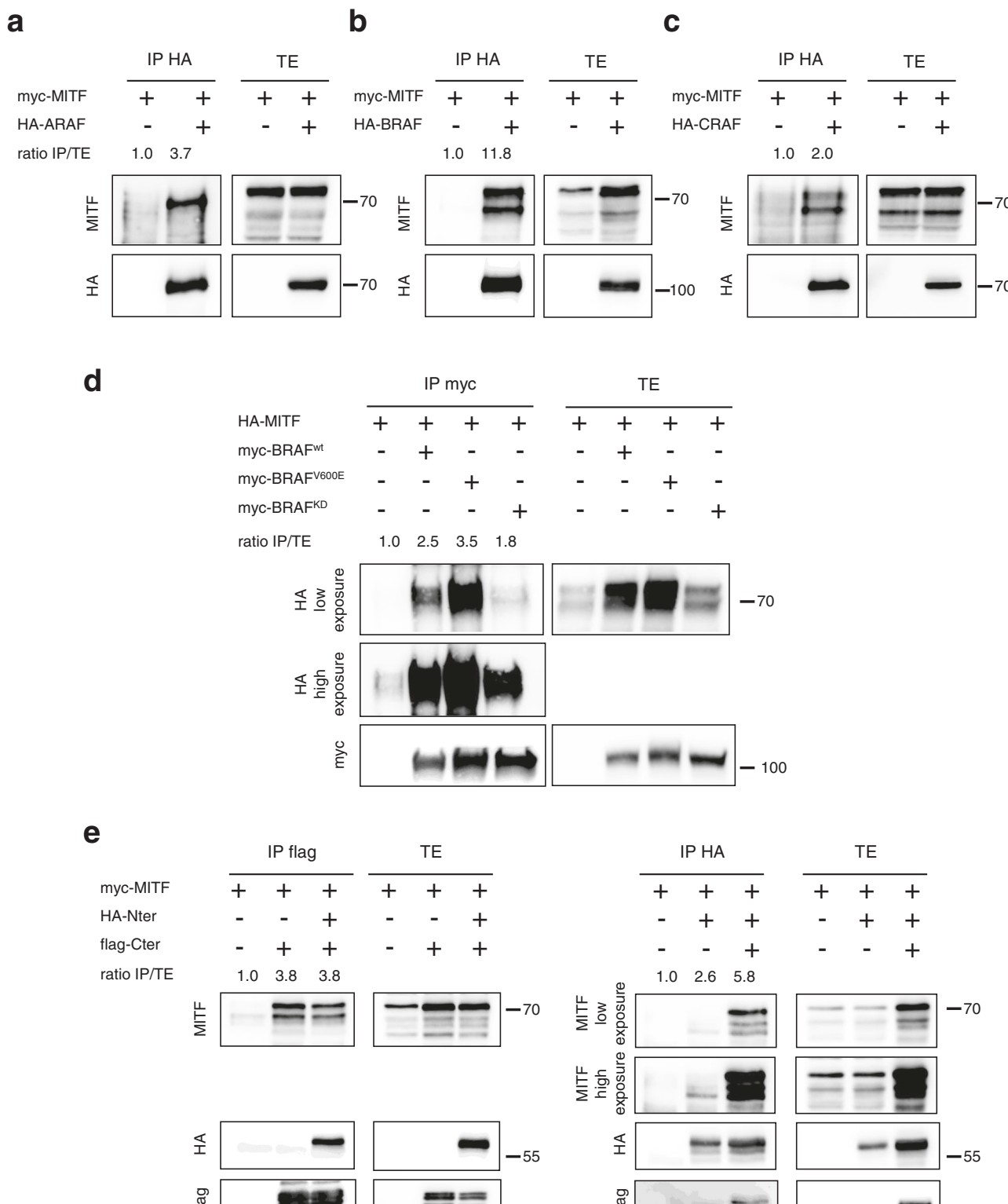

localization was analysed by immunofluorescence (Fig. 5). When expressed alone, MITF was mainly nuclear while RAF proteins displayed a clear cytoplasmic localization. However, when co-expressed, a relocalization of MITF from nucleus to cytoplasm was observed, indicating that complexes between MITF and ARAF, BRAF or CRAF, are cytoplasmic in agreement with previous observations in PLA experiments (Fig. 2b, Supplementary Fig. 4a-b). MITF shuttling was also confirmed by fractionation

experiments showing an increase of cytoplasmic MITF when coexpressed with RAF proteins (Supplementary Fig. 5). This suggests that binding to RAF proteins may retain MITF in the cytoplasm.

To better understand the functional consequence of this cytoplasmic interaction, we next investigated how RAF proteins could affect MITF transcriptional activity. HEK293T cells were transfected by constructs encoding a luciferase reporter gene

**Fig. 3 Characterization of RAF/MITF interaction.** MITF interaction with ARAF (**a**), BRAF (**b**) or CRAF (**c**). HEK293T cells were cotransfected with the myc-MITF construct and each of the three HA-ARAF, HA-BRAF, or HA-CRAF constructs (panels **a**, **b**, **c** respectively). RAF proteins were immunoprecipitated with anti-HA antibody. Immune complexes and total extracts were immunoblotted with anti-myc or anti-HA antibodies. **d** HEK293T cells are cotransfected with the HA-MITF construct and each of the three myc-BRAF$^{WT}$, myc-BRAF$^{V600E}$ or myc-BRAF$^{KD}$ constructs. Cell lysates were immunoprecipitated with anti-myc antibody. Immune complexes and total extracts were revealed with anti-HA and anti-myc antibodies. **e** HEK293T cells were cotransfected with the myc-MITF construct and HA-Nter or flag-Cter or both constructs. Cell lysates were immunoprecipitated either with anti-flag or anti-HA antibodies, and immune complexes were revealed with anti-MITF, anti-HA or anti-flag antibodies. Transfection efficiency was monitored by direct western blotting of total protein extracts. Coimmunoprecipitations were quantified using Image J software. The ratio of immunoprecipitated MITF over total MITF (IP/TE) was obtained by dividing the measured MITF signal intensity in immunoprecipitation (IP) by the MITF signal in the total extract (TE) for each condition and the ratio was set to 1 for the control condition. Coimmunoprecipitations are representative of at least three independent experiments.

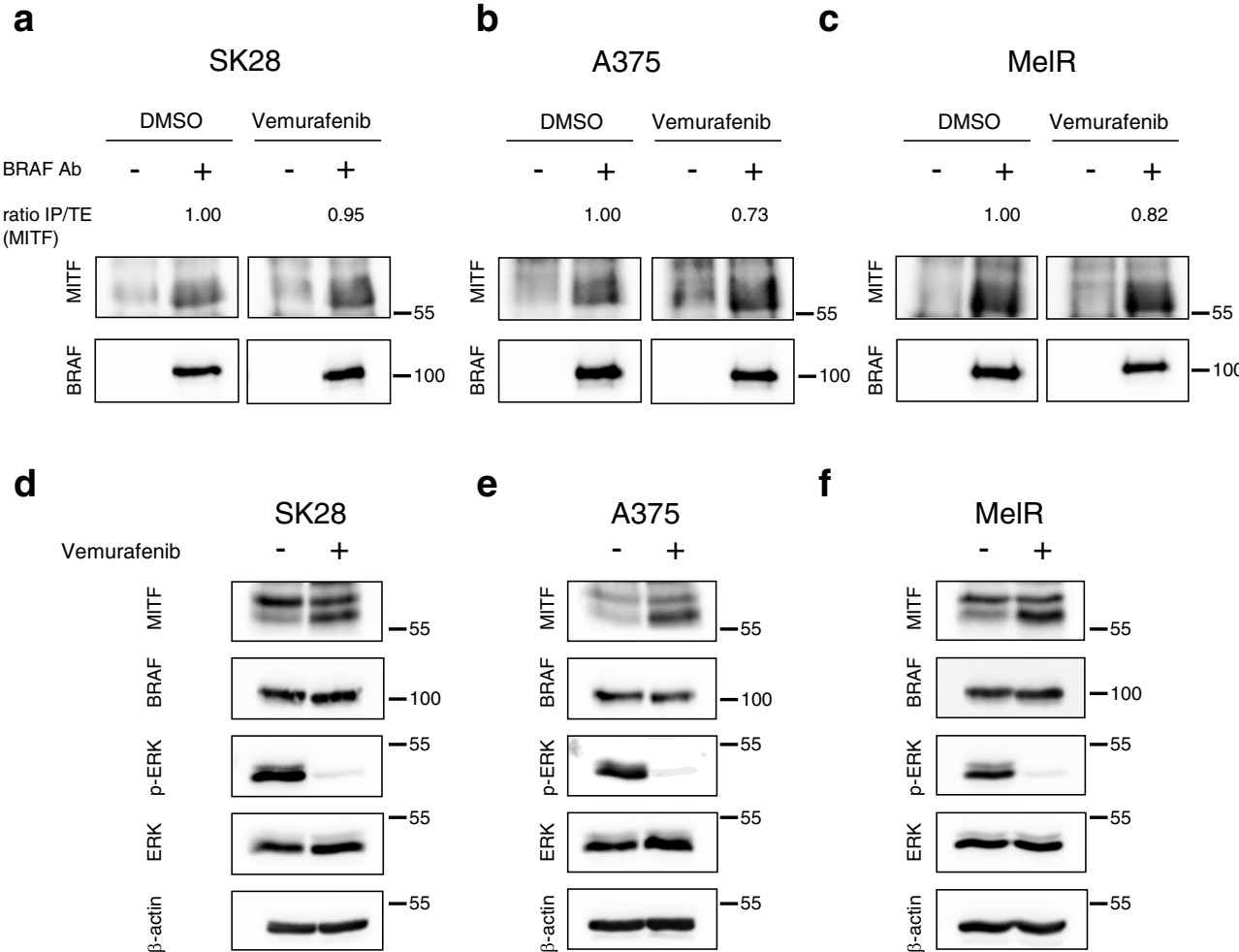

**Fig. 4 Identification of endogenous BRAF/MITF complexes in human melanoma cells.** Endogenous BRAF/MITF interaction in BRAF-mutated human melanoma cells. BRAF was immunoprecipitated with an anti-BRAF antibody in SK28 (**a**), A375 (**b**) and MelR (**c**) treated overnight with 1 μM Vemurafenib or DMSO. Immune complexes were blotted with anti-MITF and anti-BRAF antibodies. Western blot analysis of MITF and BRAF protein expression and ERK activation (pERK) in SK28 (**d**), A375 (**e**) and MelR (**f**) melanoma cells after treatment with 1 μM Vemurafenib or DMSO. Total ERK and β-actin are used as loading controls. The immunoprecipitated MITF over total MITF ratio (IP/TE(MITF)) was obtained by dividing the MITF signal intensity in immunoprecipitation by the MITF signal in the total extract for each condition. The IP/TE ratio was set to 1 in the DMSO control condition for each cell lines. Coimmunoprecipitations are representative of at least three independent experiments.

under the control of the MITF-regulated tyrosinase promoter, together with a constant amount of MITF plasmid and increasing amounts of plasmids encoding the RAF proteins (Fig. 6). Expression of MITF and RAF kinases was checked (Supplementary Fig. 6). We found that RAF kinases overexpression led to a decrease in MITF transcriptional activity, in a dose-dependent manner. Both BRAF and CRAF overexpression strongly suppressed MITF transcriptional activity while ARAF, which

possesses a weaker kinase activity, reduces MITF activity to a lesser extent (Fig. 6). Thus, the inhibition of MITF transcriptional activity by RAF proteins appears to be correlated with their ability to interact with MITF. Accordingly, the luciferase activity was highly decreased by BRAF$^{V600E}$ mutant, the strongest interactor, as compared to wild type BRAF while the BRAF kinase-dead mutant, the weakest interactor, had no effect on MITF transcriptional activity. These results showed that binding to RAF kinases

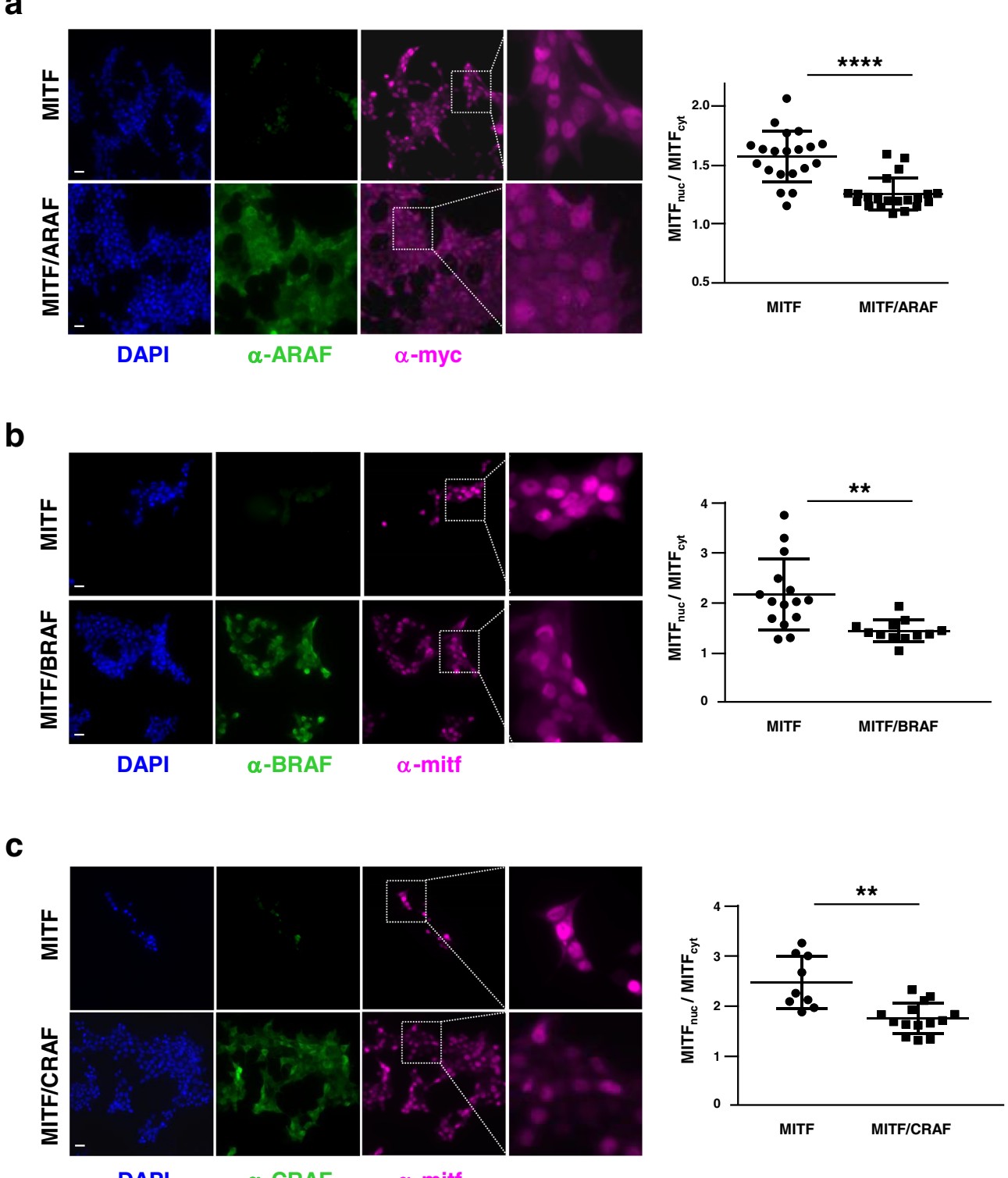

**Fig. 5 Effect of RAF proteins on MITF subcellular localization.** Subcellular localization of MITF in the presence of ARAF (**a**), BRAF (**b**) or CRAF (**c**). HEK293T cells were cotransfected with myc-MITF and HA-ARAF, HA-BRAF or HA-CRAF. Cell immunostaining was performed with anti-myc or anti-MITF and anti-ARAF, anti-BRAF or anti-CRAF. Scatter plots represent the ratio $MITF_{nuc}/MITF_{cyt}$ calculated by measuring the quantity of nuclear MITF over cytoplasmic MITF. Means with standard deviations are shown. Scale bar: 200 μm; **** $p$ value < 0.0001, ** $p$ value = 0.014 or 0.029 (for BRAF and CRAF, respectively) compared by unpaired $t$ test with Welch's correction.

negatively regulates MITF transcriptional activity. Taken together, these observations suggest a link between the intrinsic activating properties of RAF proteins, their ability to form complex with MITF and inhibition of MITF transcriptional activity.

MITF plays a critical role in melanoma cells homeostasis, acting as a master regulator of transcription of numerous target genes involved in a large panel of biological functions (proliferation, cell cycle control, survival, invasion, DNA repair,

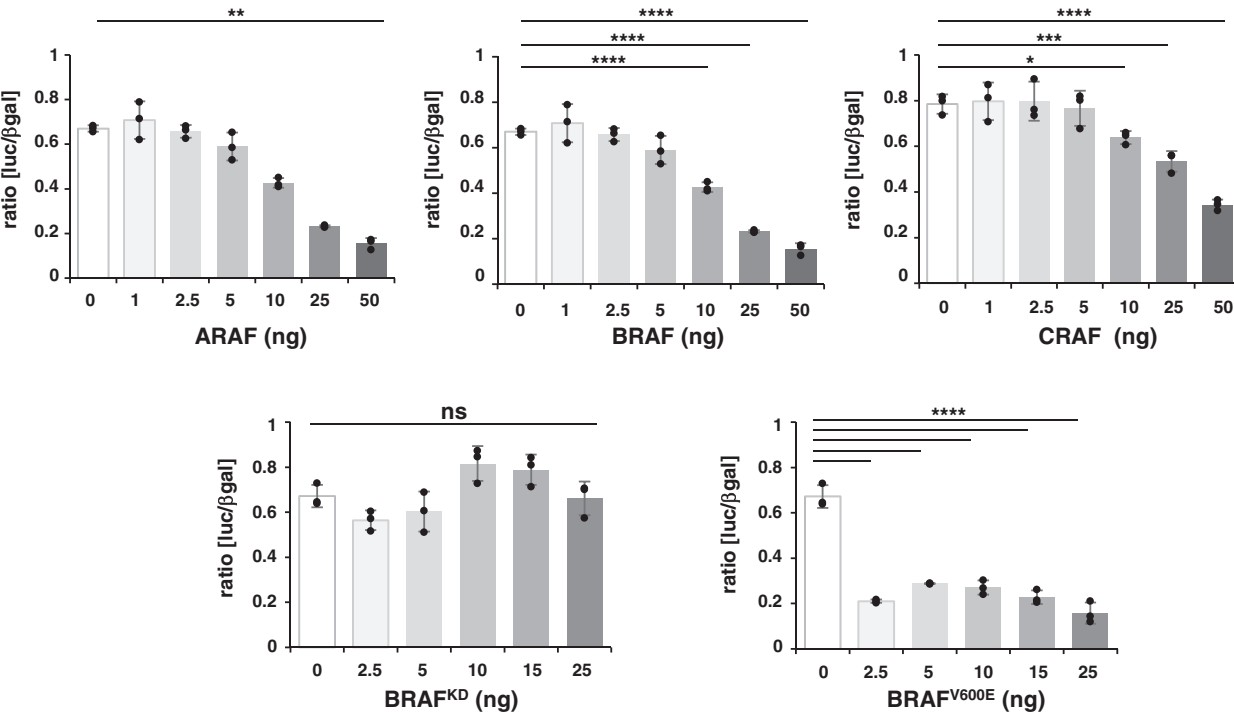

**Fig. 6 Effect of RAF proteins on MITF transcriptional activity.** HEK293T cells were cotransfected with 5 ng of MITF plasmid and increasing doses of either ARAF, CRAF, BRAF$^{WT}$, BRAF$^{V600E}$ or BRAF$^{KD}$ constructs in the presence of a TYR-Luc luciferase reporter and a control β-galactosidase reporter. The ratio of luciferase to β-galactosidase activities is shown as the mean with standard deviations of three replicates ($n = 3$). One-way ANOVA test was performed to compare all conditions and Dunnett's tests for the multiple comparisons to the no MITF condition (n.s., non-significant; * $p$ value < 0.05; ** $p$ value < 0.01; *** $p$ value < 0.001; **** $p$ value < 0.0001).

metabolism, autophagy, etc)[15,40–48]. The contradictory observations regarding the role of MITF in proliferation lead to the establishment of the MITF rheostat model whereby MITF activity is linked to melanoma cell phenotype: in this model, high levels of MITF are associated with pro-proliferative phenotype whereas lower levels are correlated with invasiveness (Fig. 7, left part)[15,42]. Recently, the MITF rheostat model was refined by incorporating six phenotypic states ranging from hyper- to under-differentiated, and associated with different level of MITF activity[49,50]. The modulation of MITF activity in melanoma cells is highly complex and partly due to a dynamic regulation at transcriptional and post-translational levels[11]. Here, we revealed an unsuspected mechanism of MITF activity modulation by demonstrating a direct interaction between RAF kinases and MITF. Indeed, overexpression of ARAF, BRAF or CRAF kinases triggers a partial subcellular relocalization of MITF in the cytoplasm, thus enabling to reduce nuclear concentrations of MITF that could fine tune MITF activity and, thus impacting phenotype switching (Fig. 7). This mechanism of proliferation regulation could happen more particularly when the pro-proliferative ERK/MAPK pathway is highly activated. Nevertheless, BRAF activity inhibition by Vemurafenib did not prevent MITF binding. This suggests that MITF binding to RAF kinases is rather due to a specific conformation than to a high kinase activity, in agreement with MITF not being a direct RAF kinase substrate. This study clearly establishes a cytoplasmic and direct binding between MITF and RAF proteins. However, the MITF transcription factor and the RAS/RAF/MEK/ERK pathway are two tightly interconnected players in melanoma, the regulation of MITF activity also involving phosphorylation of MITF by ERK[19,20,22,23]. The specific contribution of the novel regulatory mechanism by RAF/MITF complexes uncovered in our study is difficult to decipher without specific tools, such as compounds or peptides that abrogate complex formation without affecting MITF phosphorylation by ERK. Even if the biological significance of the interaction between MITF and RAF kinases deserves further investigations, our study reveals that the regulation of MITF activity by the MAPK/ERK pathway appears more complex than previously anticipated.

## Material and methods

**Cell lines**. Wild type, ARAF-only and control cells were obtained from previously described NRAS-mutated murine melanoma[7]. NRAS-mutated murine melanoma cells, named "wild type" cells in the manuscript, display normal levels of ARAF, BRAF and CRAF. ARAF-only cells and control cells are derived from these NRAS-mutated murine melanoma cells. ARAF-only cells are double knockout for BRAF and CRAF. Control cells display normal levels of BRAF and CRAF and stably express a shRNA against ARAF (TRCN0000294819, Mission shRNA library, Sigma)[7]. Cells were cultured in HAM F-12 Medium (Gibco) containing 10% fetal bovine serum (FBS), 100 mg/mL streptomycin, 100 U/mL penicillin and 2 mM L-glutamine (Invitrogen). Human embryonic kidney 293 T (HEK293T) cells were maintained in Dulbecco's modified Eagle's medium (DMEM, Gibco) supplemented with 10% FBS, 100 mg/mL streptomycin, 100 U/mL penicillin, and 1 mg/mL amphotericin B. The A375 and SK28 human melanoma cell lines were a gift from Nicolas Dumaz (Saint Louis Hospital, France) and MelR melanoma cells a gift from Caroline Robert (Gustave Roussy Hospital, France). A375 and MelR cell lines were cultured in DMEM and SK28 in RPMI medium (GIBCO) supplemented with 10% FBS, 100 U/mL penicillin and 100 mg/mL streptomycin. When indicated, cells were treated overnight with DMSO or Vemurafenib (PLX4032) (1 μM, Selleckchem). Cells were tested for mycoplasma contamination and cultured at 37 °C and 5% $CO_2$.

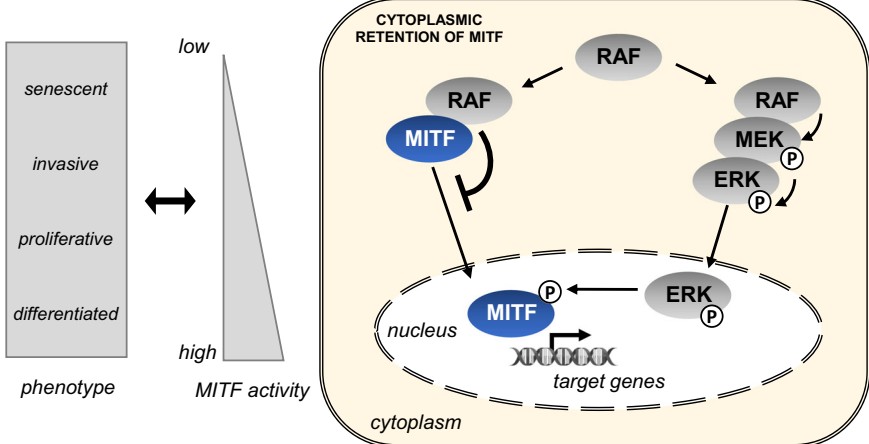

**Fig. 7 Hypothetical model. Scheme illustrating the regulation of MITF activity by direct binding with RAF kinases.** Model illustrating the regulation of MITF activity by direct binding with RAF kinases.

**Proteomics and mass spectrometry analysis**. ARAF-only or control cells were lysed in NP40 buffer (Tris pH7.5, 50 mM, NaCl, 150 mM, 0.5% NP40, protease and phosphatase inhibitors). Five or four biological replicates were prepared for each condition, respectively. Endogenous ARAF was immunoprecipitated with ARAF antibody (75804, Cell Signaling) and Pierce™ protein-A magnetic beads on 1 mg of total protein extracts. Immunoprecipitation was performed at 4 °C for 4 hours and pellets were washed 3 times in NP40 buffer and twice in 100 μL of ABC buffer (25 mM NH$_4$HCO$_3$). Beads were resuspended in ABC buffer and digested with 0.20 μg of trypsine/LysC (Promega) for 1 hour at 37 °C. Samples were loaded onto homemade Tips packed with Empore™ C18 Extraction Disks (3 M™ Discs 2215) for desalting. Peptides were eluted using 40/60 MeCN/H$_2$O + 0.1% formic acid and vacuum concentrated to dryness. Liquid chromatography was performed with an RSLCnano system (Ultimate 3000, Thermo Scientific) coupled to a Q Exactive HF-X with a Nanospay Flex ion source (Thermo Scientific). Peptides were trapped on a C18 column (75 μm inner diameter × 2 cm; nanoViper Acclaim PepMapTM 100, Thermo Scientific) with buffer A (2/98 MeCN/H$_2$O in 0.1% formic acid) at a flow rate of 2.5 μL/min over 4 min. Separation was performed on a 50 cm×75 μm C18 column (nanoViper Acclaim PepMapTM RSLC, 2 μm, 100 Å, Thermo Scientific) regulated at 50 °C with a linear gradient of 2–30% buffer B (100% MeCN in 0.1% formic acid) at a flow rate of 300 nL/min over 91 min. MS full scans were performed in the ultrahigh-field Orbitrap mass analyser in ranges $m/z$ 375–1,500 with a resolution of 120,000 at $m/z$ 200. The top 20 intense ions were subjected to Orbitrap *via* high energy collision dissociation (HCD) activation and a resolution of 15,000 with the AGC target set to 10$^5$ ions. We selected ions with charge state from 2+ to 6+ for screening. Normalized collision energy (NCE) was set at 27 and the dynamic exclusion of 40 s. For identification, the data were searched against the *Mus Musculus* UniProt canonical database (22082017 containing 16888 sequences) using Sequest–HT through proteome discoverer (version 2.0). Enzyme specificity was set to trypsin and a maximum of two-missed cleavage sites were allowed. Oxidized methionine carbamidomethyl cysteines and N-terminal acetylation were set as variable modifications. Maximum allowed mass deviation was set to 10 ppm for monoisotopic precursor ions and 0.02 Da for MS/MS peaks. The resulting files were further processed using

myProMS v3.5[51]. FDR calculation used Percolator[52] and was set to 1 % at the peptide level for the whole study. The label free quantification was performed by peptide Extracted Ion Chromatograms (XICs) computed with MassChroQ version 2.0.1[53]. For protein quantification, XICs from proteotypic peptides shared between compared conditions (TopN matching), missed cleavages and carbamidomethyl cysteine modified peptides were used. Global MAD normalization was applied on the total signal to correct the XICs for each biological replicate. To estimate the significance of the change in protein abundance, a linear model (adjusted on peptides and biological replicates) based on two-tailed $t$ tests was performed and $p$ values were adjusted with a Benjamini–Hochberg FDR. Protein with at least nine total peptides in all replicates, a twofold enrichment and an adjusted $p$ value < 0.001 were considered significantly enriched in sample comparison. Unique proteins were considered with at least four total peptides in all replicates.

**Bioinformatics analyses**. A subset of 431 interactors (Supplementary Data 2), specifically enriched in ARAF only cells, was selected as follows: partners with a number of peptides≥9, ratio>2 and adjusted $p$ value < 0.001 and partners exclusively identified in ARAF-only cells (359 and 72 proteins, respectively). One protein (Iap, UniProt ID: P03975) has been lost during ID conversion, from UniProt to Entrez. ClusterProfiler (version 4.0.5)[54] and Pathview packages[55] (version 1.32.0) on R (version 4.1.1) were used to visualize the selected interactors involved in MAPK signaling. Process and pathway enrichment analysis and protein-protein interaction (PPI) enrichment analysis were performed by using the Metascape online tool[25] (https://metascape.org). For the process and pathway enrichment analysis, terms with a $p$ value < 0.01, a minimum count of 3, and an enrichment factor >1.5 were collected and grouped into clusters based on their membership similarities. For the PPI enrichment analysis, only physical interactions in STRING (physical score >0.132) and BioGrid were used. The resultant network contains the subset of proteins that form physical interactions with at least one other member in the list. If the network contained between 3 and 500 proteins, the Molecular Complex Detection (MCODE) algorithm was applied to identify densely connected network components. Process and pathway enrichment analysis was then applied to

each MCODE component. PPI network was visualized by using Cytoscape (version 3.8.2).

**siRNA-based functional screen**. ARAF-only cells were seeded in 96-wells plate at $8.10^3$ cells per well in HAM F-12 Medium without antibiotics. After 24 hours, cells were transfected with siRNA against the 99 putative partners of ARAF selected for the screen (mouse ON-TARGETplus siRNA, pool of 4 siRNA, Dharmacon) or siCTL (ON-TARGETplus non-targeting siRNA, pool of 4 siRNA, Dharmacon) by using DharmaFECT 3 transfection reagent. After 8 hours, medium was changed and the proliferation was followed for 72 hours by using IncuCyte®. SiRNA targeting ARAF was a pool of 4 siRNA (J-042948-05, J-042948-06, J-042948-07, J-042948-08, Dharmacon). Individual siRNA targeting MITF were from Dharmacon (si mitf#1 J-047441-05, si mitf#2 J-047441-07).

**Transfection and coimmunoprecipitation**. HEK293T were transfected with pcdna3-MITF (HA-MITF or myc-MITF, gift from C.Bertolotto) and pcdna3-RAF plasmids (HA-ARAF, HA-BRAF, HA-CRAF) or pmcef-RAF plasmids (myc-BRAF, myc-BRAF$^{V600E}$ or myc-BRAF$^{KD}$) or empty vector using Lipofectamine reagent (Invitrogen). In cotransfection experiments with N-terminal or C-terminal part of BRAF, either pcdna3-HA-Nter-BRAF and/or pcdna3-flag-Cter-BRAF[37] were transfected with pcdna3-myc-MITF or empty vectors. After 48 hours, cells were lysed in NP40 buffer and extracts were precipitated overnight at 4 °C either with anti-HA (3F10, Roche), anti-Flag (M2, Sigma) or anti-Myc (9E10, Santa Cruz) and Pierce$^{TM}$ protein-G magnetic beads. Immunoprecipitates were washed with NP40 Buffer and boiled in Laemmli's sample buffer. They were then resolved by sodium dodecyl sulfate-polyacrylamide gel electrophoresis (SDS–PAGE) and transferred onto a polyvinylidene difluoride membrane (Millipore). For endogenous immunoprecipitation in mouse melanoma cells, cell lysates were incubated overnight at 4 °C with anti-ARAF antibody (4432, Cell Signaling) and Pierce$^{TM}$ protein-A magnetic beads. For endogenous immunoprecipitation in human melanoma cells, 2 mg of protein from NP40 buffer cell lysates were incubated overnight at 4 °C with anti-BRAF antibody (sc5284, Santa Cruz) and Pierce$^{TM}$ protein-G magnetic beads. For immunoprecipitation of recombinant proteins, 15 ng of human recombinant MITF protein (Origene) were incubated overnight at 4 °C in NP40 Buffer with 75 ng of human recombinant flag-ARAF (Origene) and anti-flag magnetic beads (M2, Sigma). Coimmunoprecipitations are representative of at least three independent experiments and were quantified using Image J software as indicated in figure legends.

**Cell fractionation**. HEK293T were transfected as previously described. After 48 h, fractionation was performed as described in Suzuki et al.[56]. Briefly, $5.10^6$ cells were washed in cold PBS and resuspended in lysis buffer (PBS 1X, 1 mM orthovanadate, 0.01% Igepal). An aliquot was removed as total extract. After centrifugation, the cytoplasmic fraction was found in the supernatant and the nuclear part in the pellet. SDS at a final 1% concentration and benzonase (Sigma) were added to each fraction. The cytoplasmic MITF over total MITF ratio was obtained using Image J by dividing the ratio of cytoplasmic MITF over cytoplasmic MEK-1 by the ratio of total MITF over total MEK-1.

**Western blotting and antibodies**. For SDS–PAGE analysis, the membranes were blocked with 5 % milk in PBS Tween 20 (10 %) for 30 min at room temperature. Membranes were then probed overnight at 4 °C with the appropriated primary antibodies: anti-MITF (HPA003259, Sigma), anti-HA (3F10, Roche), anti-myc (9E10, Santa

Cruz), anti-flag (M2, Sigma), anti-ARAF (4432, Cell Signaling), anti-BRAF (sc5284, Santa Cruz), anti-CRAF (610151, BD Biosciences), anti-ERK (sc93, Santa Cruz), anti-pERK (M8159, Sigma), anti-laminA/C (10298-1-AP, Proteintech), anti-MEK1 (sc219, Santa Cruz) and anti-β-actin (A1978, Sigma) antibodies. Antigen-antibody complexes were detected by horseradish peroxidase-coupled secondary antibodies followed by enhanced chemiluminescence. Signals were acquired using a cooled-CDD camera (Fusion FX Spectra, Vilber).

**Proximity ligation assay (PLA)**. Cells were grown on glass coverslips, fixed and permeabilized. PLA (Duolink) was performed according to the manufacturer's instructions (Sigma) using antibodies against ARAF (cs4432, Cell Signaling) and MITF (ab12039, abcam), BRAF (F7, Santa Cruz) and MITF (HPA003259, Sigma), or CRAF (610151, BD Biosciences) and MITF (HPA003259, Sigma). Images were captured using a 3D Leica fluorescence microscope. The average number of dots per cell (identified by its nucleus stained with DAPI) was determined by analysing at least 148 different cells with Image J software.

**Immunofluorescence**. HEK293T cells were seeded at $3.10^5$ cells per well in 6-wells slides (Millicell, Millipore) precoated with poly-L-lysine (Sigma) and transfected with pcdna3-myc-MITF and pcdna3-RAF plasmids (HA-ARAF, HA-BRAF or HA-CRAF) or empty vector as previously described. After 48 hours, cells were fixed in 4% paraformaldehyde, blocked (0.1% Triton X-100, 10% goat serum in PBS) and stained overnight at 4 °C with anti-myc antibody (Santa Cruz) and anti-ARAF (Cell Signaling) or anti-MITF (Sigma) and anti-BRAF (Santa Cruz) or anti-CRAF (BD Biosciences). Anti-mouse Alexa Fluor 594 and anti-rabbit Alexa Fluor 488 or anti-rabbit Alexa Fluor 594 and anti-mouse Alexa Fluor 488 were used for detection. Fluoroshield with DAPI (Sigma) was used as mounting medium. Images were captured using a 3D/optigrid Leica fluorescent microscope. For quantification by using Image J software, the nuclear and cytosolic compartments were selected by applying an automatic threshold (Li Dark method) to the DAPI and FITC images. The nucleus-cytoplasm ratio was then computed by dividing the mean intensity of TexasRed2 (MITF) fluorescence extracted from nucleus region by the mean intensity from cytosolic regions obtained by subtracting DAPI from the FITC surface. The background intensity was measured on each TexasRed2 image and subtracted from the mean intensities before calculating the ratio.

**Luciferase reporter assay**. HEK293T cells seeded at $10^4$ cells per well in 96-wells plate were co-transfected by using 0.3 μL of lipofectamine in a final volume of 100 μL with 50 ng of the pTYR-Luc luciferase reporter plasmid (kindly provided by C.Bertolotto), 1.7 ng of pβgal, 5 ng of pcdna3-myc-MITF and from 0 to 25 ng of pcdna3 expression vector, empty or containing the RAF coding sequences. Dual luciferase and β-galactosidase reporter assay was performed 48 hours after transfection, using Dual-Light™ Luciferase & β-Galactosidase Reporter Gene Assay System (Invitrogen). Cells were washed with saline phosphate buffer and lysed with 15 μL/well of Dual-Light™ lysis solution. After 10 min incubation at room temperature, 25 μL/well of Buffer A were added and the luciferase activity was measured for 1 second using luminometer (TriStar[2], Berthold) after injection of 100 μL/well of Galacton-Plus® diluted 1:100 in Buffer B. After 1 h incubation in the dark, the β-galactosidase activity was measured after injection of 100 μL/well of Accelerator-II reagent for 0.5 s/well. For western blot analysis of luciferase assays, cells in 96-wells plate were lysed

in Tris pH7.5, 150 mM, NaCl, 150 mM, 0.5% NP40, 0.2%SDS, protease and phosphatase inhibitors.

**Statistics and reproducibility.** All statistical analyses were conducted using GraphPad Prism. Each assay was conducted in at least three biological replicates. The exact sample size is given in the legend of each figure. The mean ± standard deviation (SD) is displayed. Statistical analysis of PLA and immunofluorescence experiments were performed by using two-tailed unpaired $t$ tests with Welch's correction when variances were significantly different. One-way or two-way ANOVA with Dunnett's multiple comparisons test were used for IncuCyte® or luciferase assays respectively ($\alpha = 0.05$). Statistics for change in protein abundance were based on two-tailed $t$ tests with $p$ values adjusted with a Benjamini–Hochberg FDR.

**Reporting summary.** Further information on research design is available in the Nature Research Reporting Summary linked to this article.

## Data availability

The datasets produced in this study are available in the PRIDE database[57] as detailed (PXD020155). The source data for the graphs and charts in the main figures are available in Supplementary Data 4, and uncropped WB images in Supplementary Fig. 7. Any remaining information can be obtained from the corresponding author upon reasonable request

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

## Acknowledgements

We thank Caroline Robert (Gustave Roussy Hospital, France) and Nicolas Dumaz (Saint Louis Hospital, France) for human melanoma cells. This work was funded by grants from the Ligue Nationale Contre le Cancer (Equipe labellisée), Fondation ARC, Gefluc and the Société Française de Dermatologie. C.E. and L.M-O. were supported by a fellowship from the Ministère Français de l'Enseignement Supérieur, de la Recherche et de l'Innovation, and Fondation ARC. This work was also supported by "Région Ile-de-France" and Fondation pour la Recherche Médicale grants (to D.L.).

## Author contributions

C.E., A.E., C.P., and S.D. conceived the study and designed the experiments. C.E., L.M.-O., L.M., and S.D. performed and analysed the experiments. C.B. provided MITF plasmids and critically revised the manuscript. F.D. carried out the MS experimental work and D.L. supervised MS and data analysis. L.B. and C.M. wrote software for immunofluorescence quantification. C.E., A.E., C.P., and S.D. wrote the manuscript. A.E., C.P., and S.D. supervised the research.

## Competing interests

The authors declare no competing interests.
