## [Peer Review File · Communications Biology]

Reviewers' comments:

Reviewer #1 (Remarks to the Author):

This is a very interesting and well-written manuscript that identifies a direct interaction of the melanocyte lineage defining transcription factor MITF with the kinase domains of RAF kinases. This is of particular relevance as MITF expression levels and function are modulated by ERK pathway activity, which is dysregulated in probably 100% of melanoma due to alterations in BRAF, and to a lesser extent other RAF genes, NRAS and receptor tyrosine kinases. To my knowledge, this is the first study showing that MITF, a known ERK substrate, is directly binding to ERK pathway components and that RAF kinase modulate its subcellular localisation and hence activity. Very interesting is also the finding that the MITF/BRAF interaction is influenced by the activity status of the latter. The data are convincing and presented in a logical way.

I have only a few minor corrections/suggestions that could further improve this already very advanced manuscript.

1. Figure 3d and 3e. The labelling "HA low" and "HA high" is a bit misleading as it could be understood as different expression level. My understanding is that we are looking here at short and long exposures of the same Western blot membrane.

2. p.7 The authors mention the BRAFK499M mutant as kinase-dead, are they maybe referring to K483M, a substitution that is has been used in many other publications? This should be clarified.

3. Fig.5 represents an interesting finding. Could the authors correlate the decline in tyrosinase promoter::luc reporter with activity MITF expression levels?

Reviewer #2 (Remarks to the Author):

Manuscript: "New level of regulation of MITF activity by direct interaction with RAF proteins" by Charlene Estrada, Liliana Mirabal-Ortega, Florent Dingli, Laetitia Besse, Cedric Messaoudi, Damarys Loew, Celio Pouponnot, Corine Bertolotto, Alain Eychène and Sabine Druillennec.

Manuscript number: COMMSBIO-21-0906

The authors explore the role of ARAF in NRAS-driven melanoma using mass spectrometry with a siRNA-based functional screen and identify MITF along with other interactors as a binding partner of ARAF. The direct interaction between ARAF and MITF was confirmed using coimmunoprecipitation experiments and takes place in the cytoplasm of melanoma cells. Furthermore, the authors show that overexpression of ARAF, BRAF or CRAF kinases leads to relocalization of MITF into the cytoplasm, which might regulate its transcriptional activity.

Overall, the study is designed well with a clever use NRAS-driven melanoma mice and other models. The manuscript is structured and integrated well into the existing literature, building on questions that were left open in previous studies.

However, here are a few comments on points that could be improved:

Major:

1. The authors could perform pathway/network analysis on the significant binding partners of ARAF identified using mass spectrometry. This would reconfirm the existing knowledge on the expected RAS/RAF/MEK/ERK pathway and might also show other pathways apart from the expected ones. Pathway analysis can be performed with tools like Reactome or Ingenuity Pathway Analysis for example.

2. The authors could add a pathway illustration to manuscript to make it easier to follow the various steps of the pathway discussed in the manuscript.

3. The manuscript needs to have a limitations section in the discussion.

Minor:

1. In Figure 1a, the authors should mark some of the proteins of interest in the plot that they are discussing in the text.
2. The authors should make the figures more color-blind friendly where possible, for example Fig 1b.

Reviewer #3 (Remarks to the Author):

In their manuscript titled 'New level of regulation of MITF activity by direct interaction with RAF proteins', Estrada et al., show a novel interaction between ARAF and MITF in melanoma cells. They continue by validating this finding in overexpression systems, while further showing in these same systems that also BRAF and CRAF interact with overexpressed MITF. They then show in this same model that the expression of any of the RAF proteins can limit the transcriptional activity of MITF.

While the primary finding that ARAF can interact with MITF is interesting, there are some shortcomings that I will highlight below, split up in major and minor comments.

Major comments:

1. Many of the primary findings of this work were made in overexpression systems, and not validated in melanoma cells. The authors should validate their main findings (that is MITF-BRAF interaction, MITF-CRAF interaction and RAF inhibition of MITF activity) in melanoma cells proper.
2. It is unclear to me how the immunoprecipitation experiments are quantified. As an example, in Figure 3d, the authors claim that there is 2.5-fold more HA-MITF precipitated in the myc-BRAF^{WT} condition than in a condition lacking any RAF overexpression, even though this does seem correct, given that the total levels of HA-MITF also differ in the input samples. The authors should perform the quantifications again, but instead normalize to the input of each of the proteins. This should be done for all of the immunoprecipitation experiments. Extending this point, it is unclear to me how many replicates of these experiments were performed. Such quantifications can be done over multiple replicates to increase confidence in the findings. Also, again in Figure 3d, why is there an immunoprecipitation of HA-MITF in samples lacking any overexpression of HA-MITF?
3. In the immunofluorescence experiments presented in Figure 4, why can RAF signal be observed in cells lacking RAF overexpression? This makes the interpretation of the findings difficult. These experiments should also be validated in different systems, such as cellular fractionation to independently immunoblot the nuclear and cytoplasmic fractions.

Minor comments:

1. Figure 1a is difficult to comprehend, and the mass spectrometry data should be presented separately from the experimental workflow.
2. Figure 2b is difficult to comprehend, and should be visualized differently.
3. On page 5, what do the authors mean with "Control cells display normal levels of BRAF and CRAF, but express low level of ARAF shRNA-mediated knockdown, thus allowing for relative quantification of the data."? The control condition for this experiment should be carefully stated, and could perhaps be incorporated in the workflow in Figure 1a.
4. The authors claim, on the basis of several knockdown experiments, that MITF is required for ARAF-only cells growth. However, MITF is equally required for optimal growth of BRAF-driven melanoma cells. The authors should compare the relative effects of MITF knockdown in ARAF only cells to those in wildtype cells to increase the relevance of this finding.
5. To show that the kinase activity of BRAF is required for the interaction with MITF, can the BRAFⁱ vemurafenib not be used? This will also add relevance to the findings.
6. In Figure 2b and Supplementary Figure 2a, b, the outline of the cells investigated should be shown with a membrane stain.

In conclusion, while the authors present findings that are of relevance, some questions remain. Should these points be properly addressed during revisions, I feel that this manuscript would be a strong candidate for publication.

We greatly thank the reviewers for their positive reviews and constructive comments, which helped us to improve the manuscript.

Reviewers' comments:

Reviewer #1 (Remarks to the Author):

This is a very interesting and well-written manuscript that identifies a direct interaction of the melanocyte lineage defining transcription factor MITF with the kinase domains of RAF kinases. This is of particular relevance as MITF expression levels and function are modulated by ERK pathway activity, which is dysregulated in probably 100% of melanoma due to alterations in BRAF, and to a lesser extent other RAF genes, NRAS and receptor tyrosine kinases. To my knowledge, this is the first study showing that MITF, a known ERK substrate, is directly binding to ERK pathway components and that RAF kinases modulate its subcellular localisation and hence activity. Very interesting is also the finding that the MITF/BRAF interaction is influenced by the activity status of the latter. The data are convincing and presented in a logical way.

I have only a few minor corrections/suggestions that could further improve this already very advanced manuscript.

1. Figure 3d and 3e. The labelling "HA low" and "HA high" is a bit misleading as it could be understood as different expression level. My understanding is that we are looking here at short and long exposures of the same Western blot membrane.

- In **Figure 3d**, the labelling "HA low" and "HA high" has been modified by "HA low exposure" and "HA high exposure". In **Figure 3e**, the labelling "MITF low" and "MITF high" has been modified by "MITF low exposure" and "MITF high exposure".

2. p.7 The authors mention the BRAF^{K499M} mutant as kinase-dead, are they maybe referring to K483M, a substitution that is has been used in many other publications? This should be clarified.

- We thank the reviewer for this comment. We indeed used the well-known BRAF^{K483M} mutant. The previous annotation referred to the HA tagged version of the protein with the addition of 16 amino acids from the tag. We modified the text accordingly by referring to the usual name of this mutant (BRAF^{K483M}).

3. Fig.5 represents an interesting finding. Could the authors correlate the decline in tyrosinase promoter::luc reporter with activity MITF expression levels?

- We are not sure to fully understand this point. We surmise the referee asks whether it is possible to correlate the decline in tyrosinase promoter::luc reporter activity with MITF expression levels, by interchanging the words "activity" and "with".

To address this question, we performed a luciferase assay either by using escalating doses of pcDNA3-MITF plasmid (from 0 to 30ng) without BRAF or increasing doses of BRAF (from 0 to 25ng of plasmid) in presence of 5ng MITF encoding plasmid (**Supplementary Fig. 6a**). We also examined MITF and BRAF protein expression by western blot in the same experimental conditions (**Supplementary Fig. 6b**). Of note, the pcDNA3-MITF plasmid drives MITF expression thanks to a CMV promoter that responds to the MAPK pathway activation, thus explaining the increase in MITF protein level following BRAF expression. A positive correlation between MITF activity and its expression level is observed with escalating amounts of MITF in absence of BRAF while a negative correlation is found if increasing amounts of BRAF are used at a constant dose of MITF. This confirms that BRAF inhibits MITF transcriptional activity.

Reviewer #2 (Remarks to the Author):

The authors explore the role of ARAF in NRAS-driven melanoma using mass spectrometry with a siRNA-based functional screen and identify MITF along with other interactors as a binding partner of ARAF. The direct interaction between ARAF and MITF was confirmed using coimmunoprecipitation experiments and takes place in the cytoplasm of melanoma cells. Furthermore, the authors show that overexpression of ARAF, BRAF or CRAF kinases leads to relocalization of MITF into the cytoplasm, which might regulate its transcriptional activity.

Overall, the study is designed well with a clever use NRAS-driven melanoma mice and other models. The manuscript is structured and integrated well into the existing literature, building on questions that were left open in previous studies.

However, here are a few comments on points that could be improved:

Major:

1. The authors could perform pathway/network analysis on the significant binding partners of ARAF identified using mass spectrometry. This would reconfirm the existing knowledge on the expected RAS/RAF/MEK/ERK pathway and might also show other pathways apart from the expected ones. Pathway analysis can be performed with tools like Reactome or Ingenuity Pathway Analysis for example.

- We thank the reviewer for this suggestion. As recommended, we performed a KEGG pathway analysis on a subset of 431 ARAF partners found enriched in ARAF-only cells and selected on the following parameters: 359 proteins with number of peptides ≥ 9 , ratio > 2 and adjusted p-value < 0.001 and 72 partners exclusively identified in ARAF-only cells. A new figure (**Supplementary Figure 1**) has been added in the revised version confirming a MAPK signaling pathway enrichment, including its direct downstream MEK1 substrate, indicating the reliability of our experimental approach. Of note, NRAS, the direct upstream interactor of ARAF, is also found in the interactome but didn't reach all the cut-offs. While the number of peptides and ratio were correct (peptides = 13, ratio = 12), the p-value = 0.009 was above the selected cut-off. In addition, several 14-3-3 proteins (*Ywhab* and *Ywhaz* coding genes) are present in the 431 ARAF interactors subset. Although not included in the KEGG maps (**Supplementary Figure 1**), 14-3-3 proteins are also involved in MAPK signaling by directly binding and regulating RAF kinases.

We also performed process and pathway enrichment analysis as well as Protein-Protein Interaction (PPI) enrichment analysis with Metascape. These analyses revealed an enrichment in Rho GTPases signaling and mitochondrial processes, such as TCA cycle and respiratory electron transport, mitochondrial translation or fatty acid beta-oxidation. These data are now included in a new table (**Table 2**) and a new figure (**Supplementary Figure 2**). They are of potential interest since previous works reported the involvement of CRAF in Rho signalling (Niault et al., J Cell Biol, 2009 ; Ehrenreiter et al., J Cell Biol, 2005) and RAF kinases in mitochondrial functions (Wang et al., PNAS, 1996 ; Wang et al., Cell, 1996 ; O'Neill et al., Science, 2004 ; Rauch et al., Cell Death and Differentiation, 2016) with the activated form of BRAF found localized in the mitochondria where it regulates oxidative metabolism (Lee et al., The Journal of Clinical Endocrinology & Metabolism , 2011). This is discussed in the revised version of the manuscript.

2. The authors could add a pathway illustration to manuscript to make it easier to follow the various steps of the pathway discussed in the manuscript.

- As suggested, a model illustrating the regulation of MITF activity by direct binding with RAF kinases has been added in **Figure 7**.

3. The manuscript needs to have a limitations section in the discussion.

- As suggested, limitations of the present work are now included at the end of the main text where the model is discussed (**Figure 7**).

Minor:

1. In Figure 1a, the authors should mark some of the proteins of interest in the plot that they are discussing in the text.

- As suggested, the 16 proteins found to display anti- or pro-proliferative properties in the functional siRNA-based screen are highlighted on the volcano plot in **Figure 1a**.

2. The authors should make the figures more color-blind friendly where possible, for example Fig 1b.

- As recommended by the reviewer, colour combinations used in the different figures have been modified. In particular, the red and green contrast has been replaced by red and blue, green and magenta or red and black contrasts in **Figures 1 and 5 and Supplementary Figures 3**. These colours have been chosen according to the submission guidelines from Communications Biology.

Reviewer #3 (Remarks to the Author):

In their manuscript titled 'New level of regulation of MITF activity by direct interaction with RAF proteins', Estrada et al., show a novel interaction between ARAF and MITF in melanoma cells. They continue by validating this finding in overexpression systems, while further showing in these same systems that also BRAF and CRAF interact with overexpressed MITF. They then show in this same model that the expression of any of the RAF proteins can limit the transcriptional activity of MITF.

While the primary finding that ARAF can interact with MITF is interesting, there are some shortcomings that I will highlight below, split up in major and minor comments.

Major comments:

1. Many of the primary findings of this work were made in overexpression systems, and not validated in melanoma cells. The authors should validate their main findings (that is MITF-BRAF interaction, MITF-CRAF interaction and RAF inhibition of MITF activity) in melanoma cells proper.

- In the former version of the paper, we observed MITF interaction with the three members of RAF kinase family not only by overexpression in HEK293T (**Figure 3**) but also at endogenous levels in mouse melanoma cells (**Figure 1, Figure 2 and new Supplementary Figure 4** corresponding to **previous Supplementary Figure 2**). Indeed, the endogenous ARAF/MITF complex was observed in mouse melanoma cells, firstly by proteomic experiments (**Figure 1**) and then confirmed in coimmunoprecipitation and PLA experiments (**Figure 2**). MITF complexes with BRAF and CRAF were also demonstrated endogenously in mouse melanoma cells by PLA experiments (**New Supplementary Figure 4** corresponding to previous **Supplementary Figure 2**).

- In the revised version, we further validated the endogenous interaction of BRAF^{V600E} with MITF in three human melanoma cells (**new Figure 4**). After immunoprecipitation of BRAF, we were able to detect MITF binding in SK28, A375 and MeIR human melanoma cells, thus confirming our previous observations at the endogenous level in human cells, as requested by the referee.

- Evaluation of the endogenous inhibition of MITF activity by RAF proteins in human melanoma cells represents a highly challenging task. At the endogenous level, without overexpression, modulation of the MITF/RAF complex can be achieved through silencing or chemical inhibition of RAF proteins. However, these two approaches cannot be successful for this specific question. As we previously

reported in Dorard et al. (2017), compensatory effects between the different RAF kinases prevent the evaluation of their specific role and the complete silencing of RAF kinases is incompatible with melanoma cells survival. On the other hand, the impact of chemical RAF inhibition on MITF activity is difficult to interpret since, as previously reported, these compounds alter MAPK activation and, therefore, interfere with the regulation of MITF by ERK. To study the specific impact of RAF on MITF, independently of ERK activation, compounds or peptides preventing RAF/MITF interaction without impacting ERK activation are necessary. To reach this goal a complete understanding of complex formation is required and represents a work on itself that is out of the scope of this manuscript.

2. It is unclear to me how the immunoprecipitation experiments are quantified. As an example, in Figure 3d, the authors claim that there is 2.5-fold more HA-MITF precipitated in the myc-BRAF^{WT} condition than in a condition lacking any RAF overexpression, even though this does seem correct, given that the total levels of HA-MITF also differ in the input samples. The authors should perform the quantifications again, but instead normalize to the input of each of the proteins. This should be done for all of the immunoprecipitation experiments. Extending this point, it is unclear to me how many replicates of these experiments were performed. Such quantifications can be done over multiple replicates to increase confidence in the findings. Also, again in Figure 3d, why is there an immunoprecipitation of HA-MITF in samples lacking any overexpression of HA-MITF?

- We agree with the reviewer that the quantification method for coimmunoprecipitation experiments was not clearly explained. In fact, quantifications were exactly performed as requested by the reviewer. In all figures, the IP signal was corrected by normalization on the input signal (TE). For example, in **Figure 3d**, we measured an 8-fold increase of MITF IP signal and a 3.2-fold increase of MITF TE signal in BRAF^{WT} condition compared to control. The 2.5 ratio IP/TE indicated in the **Figure 3d** is obtained by dividing 8 by 3.2.

To clarify the quantification method, we modified the legends of **Figure 3** and **new Figure 4**. In **Figure 3** legend, we modified the sentence “*Coimmunoprecipitations were quantified using Image J software by measuring signal intensity in immunoprecipitation over total extract and normalized on control lane.*” by “*The ratio of immunoprecipitated MITF over total MITF (IP/TE) was obtained by dividing the measured MITF signal intensity in immunoprecipitation (IP) by the MITF signal in the total extract (TE) for each condition and the ratio was set to 1 for the control condition.*”. In **Figure 4** legend, we indicated that: “*The immunoprecipitated MITF over total MITF ratio (IP/TE(MITF)) was obtained by dividing the MITF signal intensity in immunoprecipitation by the MITF signal in the total extract for each condition. The IP/TE ratio was set to 1 in the DMSO control condition for each cell lines.*”.

- As indicated in the Material and Methods section, “*coimmunoprecipitations are representative of at least three independent experiments*”. We added this comment in **Figure 3** and **new Figure 4** legends.

- In **Figure 3d**, there is no sample “lacking any overexpression of HA-MITF”. As indicated in **Figure 3d** labelling (+), HA-MITF is expressed in all conditions (see the four lanes of TE). The very weak signal observed in the IP control condition by anti-myc antibody is a non-specific background very often observed in coimmunoprecipitations. This also enables quantification.

3. In the immunofluorescence experiments presented in Figure 4, why can RAF signal be observed in cells lacking RAF overexpression? This makes the interpretation of the findings difficult. These experiments should also be validated in different systems, such as cellular fractionation to independently immunoblot the nuclear and cytoplasmic fractions.

- In the new **Figure 5** corresponding to **Figure 4** of the former version of the manuscript, the very low signal observed in few cells lacking RAF overexpression is a background signal due mainly to the secondary antibody as often observed in immunofluorescence experiments. In contrast, a cytoplasmic signal is observed in cells transfected by RAF constructs.

- As requested by the referee, the relocation of MITF from the nucleus to the cytoplasm in presence of RAF proteins was confirmed by fractionation experiments. In the **new Supplementary Figure 5**, we

show that ARAF, BRAF and CRAF increase the amount of cytoplasmic MITF in agreement with the immunofluorescence experiments. The quantification of cytoplasmic MITF compared to total MITF was obtained by dividing the ratio of cytoplasmic MITF over cytoplasmic MEK-1 by the ratio of total MITF over total MEK-1. The quantification method is included in the Material and Methods section and the figure legend.

Minor comments:

1. Figure 1a is difficult to comprehend, and the mass spectrometry data should be presented separately from the experimental workflow.

- **Figure 1** has been reorganized. The volcano plot is presented separately in panel **a**. Two new Supplementary **Figures 1 and 2** related to **Figure 1** have been added as suggested by the reviewer #2. Data from panel **b** are now included in the new **Table 3**.

2. Figure 2b is difficult to comprehend, and should be visualized differently.

- As suggested by the reviewer, we added labels to clarify the PLA figures (**Figure 2 and Supplementary Figure 4**). Notably, we entitled each figure to explain which complex are visualized by PLA. Explicit labelling of cell lines used for each experiment was also added. Altogether, these labels help to better understand how experiments were conducted. PLA quantifications were performed using several microscopic fields, containing hundreds of cells. PLA images are conventional PLA illustrations, as previously published (Dorard et al., 2017), showing a representative single cell chosen among all quantified cells and illustrating coherently PLA quantification.

3. On page 5, what do the authors mean with "Control cells display normal levels of BRAF and CRAF, but express low level of ARAF shRNA-mediated knockdown, thus allowing for relative quantification of the data."? The control condition for this experiment should be carefully stated, and could perhaps be incorporated in the workflow in Figure 1a.

- Control cells express endogenous CRAF and BRAF but are knock-down (KD) for ARAF thanks to the stable expression of an ARAF-targeted shRNA. Consequently, they display normal levels of BRAF and CRAF but low levels of ARAF, thus enabling relative quantification of proteomic data. As suggested, we added a schematic representation of RAF protein contents in control and ARAF only cells in **Figure 1a**. In the Material and Methods – Cell lines section, we clearly mention : *“Wild type, ARAF-only and control cells were obtained from previously described NRAS-mutated murine melanoma. NRAS-mutated murine melanoma cells, named “wild type” cells in the paper, display normal levels of ARAF, BRAF and CRAF. ARAF-only cells and control cells are derived from these NRAS-mutated murine melanoma cells. ARAF-only cells are double knockout for BRAF and CRAF. Control cells display normal levels of BRAF and CRAF and stably express a shRNA against ARAF”*.

4. The authors claim, on the basis of several knockdown experiments, that MITF is required for ARAF-only cells growth. However, MITF is equally required for optimal growth of BRAF-driven melanoma cells. The authors should compare the relative effects of MITF knockdown in ARAF only cells to those in wildtype cells to increase the relevance of this finding.

- As suggested by the reviewer, we evaluated the effect of MITF knockdown on the growth of wild type cells. Two panels were added to the new Supplementary **Figure 4a and b**. We confirmed the proliferative effect of MITF in murine melanoma cells expressing normal levels of ARAF, BRAF and CRAF by using two distinct siRNA against MITF in comparison to control siRNA. The extinction level of MITF expression was controlled by western blot. Of note, knockdown of ARAF has a very weak effect on wild type cell proliferation as we already published.

5. To show that the kinase activity of BRAF is required for the interaction with MITF, can the BRAFi vemurafenib not be used? This will also add relevance to the findings.

- Since a correlation between the intrinsic activating properties of RAF kinases and their binding to MITF was observed, cells were treated with Vemurafenib, as suggested by the reviewer. This molecule is a widely known BRAF^{V600E} inhibitor used in clinics. It acts as an ATP-competitive kinase inhibitor that blocks the active site of the kinase. Vemurafenib treatment was performed in BRAF-mutated human melanoma cells since Vemurafenib induces a paradoxical activation of ERK in NRAS-mutated cells as already described, but also in HEK293T cells surexpressing wild type forms of RAF proteins. In the new **Figure 4b**, it appears that as expected, Vemurafenib treatment inhibits MAPK pathway activation. In these conditions, we observed only a slight decrease in the ability of MITF to interact with mutated BRAF compared to untreated cells. This indicates that the active site of RAF kinases is not the MITF binding domain and that the interaction does not require kinase activity. The lack of kinase activity requirement is also supported by results in **Fig. 3e** (left panel) showing that the N-terminus, known to decrease C-terminus kinase activity (Cutler, et al., 1998 ; Chong,et al., 2003 ; Hmitou, et al., 2007), did not modify complex formation between the C-terminus and MITF. Accordingly, no phosphorylation sites or consensus for phosphorylation by RAF kinases indicating that MITF could be a direct RAF substrate have been reported. This also suggests that the ability of RAF kinases to bind MITF is not linked to a fully functional kinase active site, but rather due to conformational aspects.

6. In Figure 2b and Supplementary Figure 2a, b, the outline of the cells investigated should be shown with a membrane stain.

- In line with minor point #2, PLA quantification consists in counting the number of PLA foci and the number of nuclei stained with DAPI in a microscopic field. Therefore, the ratio foci/cell corresponds to the mean number of complex per cell identified by its nucleus. In this case, PLA experiments do not require membrane staining since the purpose of this experiment is to validate a given interaction.

In conclusion, while the authors present findings that are of relevance, some questions remain. Should these points be properly addressed during revisions, I feel that this manuscript would be a strong candidate for publication.

REVIEWERS' COMMENTS:

Reviewer #1 (Remarks to the Author):

The authors have successfully addressed all my comments.

Reviewer #2 (Remarks to the Author):

The authors addressed all points raised by this reviewer. I recommend the manuscript for publication.

Reviewer #3 (Remarks to the Author):

In the revised version of their manuscript 'New level of regulation of MITF activity by direct interaction with RAF proteins', Estrada et al., have (more than) adequately addressed the concerns I raised. I therefore fully support publication of their manuscript in Communications Biology.